# Continuous Movement Monitoring at Home Through Wearable Devices: A Systematic Review

**DOI:** 10.3390/s25164889

**Published:** 2025-08-08

**Authors:** Gianmatteo Farabolini, Nicolò Baldini, Alessandro Pagano, Elisa Andrenelli, Lucia Pepa, Giovanni Morone, Maria Gabriella Ceravolo, Marianna Capecci

**Affiliations:** 1Department of Experimental and Clinical Medicine, Politecnica delle Marche University, 60126 Ancona, Italy; g.farabolini@univpm.it (G.F.); nicolo.baldini@pm.univpm.it (N.B.); a.pagano@pm.univpm.it (A.P.); e.andrenelli@univpm.it (E.A.); m.g.ceravolo@univpm.it (M.G.C.); 2Department of Information Engineering, Politecnica delle Marche University, 60131 Ancona, Italy; l.pepa@univpm.it; 3Department of Clinical Medicine, Public Health, Life and Environmental Sciences, University of L’Aquila, 67100 L’Aquila, Italy; giovanni.morone@uniaq.it

**Keywords:** wearable sensors, home-based monitoring, motor symptoms, inertial measurement units (IMUs), remote assessment, digital health

## Abstract

**Highlights:**

**What are the main findings?**
Wearable sensors—especially IMUs, accelerometers, and gyroscopes—are widely used for continuous home-based motor monitoring, particularly in neurological conditions like Parkinson’s disease.Most studies report high feasibility and patient compliance (≥70%), but only 5.6% were randomized trials, limiting the strength of clinical recommendations.

**What are the implications of the main findings?**
Wearable devices are reliable tools for the real-world assessment of motor symptoms, potentially complementing traditional in-clinic evaluations.Broader clinical adoption will require overcoming challenges such as clinician awareness, standardization, data privacy, and equitable access to technology.

**Abstract:**

Background: Wearable sensors are a promising tool for the remote, continuous monitoring of motor symptoms and physical activity, especially in individuals with neurological or chronic conditions. Despite many experimental trials, clinical adoption remains limited. A major barrier is the lack of awareness and confidence among healthcare professionals in these technologies. Methods: This systematic review analyzed the use of wearable sensors for continuous motor monitoring at home, focusing on their purpose, type, feasibility, and effectiveness in neurological, musculoskeletal, or rheumatologic conditions. This review followed PRISMA guidelines and included studies from PubMed, Scopus, and Web of Science. Results: Seventy-two studies with 7949 participants met inclusion criteria. Neurological disorders, particularly Parkinson’s disease, were the most frequently studied. Common sensors included inertial measurement units (IMUs), accelerometers, and gyroscopes, often integrated into medical devices, smartwatches, or smartphones. Monitoring periods ranged from 24 h to over two years. Feasibility studies showed high patient compliance (≥70%) and good acceptance, with strong agreement with clinical assessments. However, only half of the studies were controlled trials, and just 5.6% were randomized. Conclusions: Wearable sensors offer strong potential for real-world motor function monitoring. Yet, challenges persist, including ethical issues, data privacy, standardization, and healthcare access. Artificial intelligence integration may boost predictive accuracy and personalized care.

## 1. Introduction

Non-communicable diseases (NCDs) represent a significant issue from various perspectives, including healthcare, socioeconomics, and politics [1]. The World Health Organization (WHO) is vigorously fighting to reduce the high burden of NCDs: the 2030 Agenda for Sustainable Development identified NCDs as a “major challenge for sustainable development” [2]. In addition, life expectancy and age-standardized disease burden are expected to increase globally between 2022 and 2050: the impact of NCDs on this burden will be increasingly greater than the impact of communicable, maternal, neonatal, and nutritional diseases (CMNNs). Therefore, years lived with disability (YLDs) and disability-adjusted life years (DALYs) are expected to increase worldwide in the coming years [3]: many people will need healthcare assistance at a time when health systems are suffering from a lack of economic and human resources. The expanded Care Model stated that home assistance should be considered a core feature in patients’ management, especially in the latest disease stages [4]. This goal requires a solid network between health professionals, patients, and caregivers. The constant interchange of information is crucial, and the continuous monitoring of health conditions is necessary to ensure a safe homestay in stable clinical conditions.

Nowadays, continuous home monitoring is an innovative and growing research field.

Technology offers a wide range of solutions for patient monitoring from smart homes, characterized by stand-alone devices and computers with or without wearable sensors [5,6], to light portable sensor devices for detecting health signals such as the following: pulse rate [7,8], blood pressure [9], or ECG [10]; blood glucose levels [11]; blood oxygen saturation or respiratory rate [12,13]; and neurological signals for the detection of seizures [14], sleep disorders [15], or movement disorders [16]. Wearable devices play a crucial role in monitoring, given the importance of physical activity, gait, and upper limb movements. Motor functioning can greatly affect the quality of life, independence in activities of daily living, and instrumental activities [17,18]. Motor activity can represent a predictor of health and survival [18,19], and the WHO highlighted the importance of the amount and quality of physical activity in the prevention and management of NCDs [20].

After the COVID-19 pandemic, many papers were published regarding wearable devices for detecting movement, such as inertial measurement units (IMUs). These devices employ gyroscopes, magnetometers, or accelerometers, alone or combined, to measure the body’s accelerations, angular velocities, and orientation according to the earth’s magnetic axis [21]. Nowadays, IMUs are embedded in common accessories or clothing such as T-shirts, watches, glasses, and smartphones [21]. IMUs register and analyze data using algorithms that are becoming increasingly sophisticated thanks to the expansion of artificial intelligence and provide feedback to the subject on the amount of physical activity, sleep, energy expenditure, posture, limb movement, etc. [22,23,24]. Picerno et al. [21] identified several emerging applications: mobile health solutions (e.g., for assessing frailty, fall risk, chronic neurological diseases, and monitoring and promoting active living), occupational ergonomics, rehabilitation and telerehabilitation, and cognitive assessment. They emphasized that over the past five years, wearable IMUs have shifted from being predominantly used in laboratory environments to unsupervised, naturalistic settings [25,26,27,28]. This shift has opened numerous opportunities for IMU-based applications in mHealth, occupational settings, telerehabilitation [29], and psychophysics, where wearable IMUs demonstrated their potential in assessing perceptual and cognitive abilities [30,31].

Although IMUs are the most widely used monitoring technology, they are not widespread. In an Italian survey, they were used by less than 18% of clinicians [32], and mainly for clinical research, clinicians needed more training in terms of evidence of their effectiveness, technology applications, application protocols, indications for use, and contraindications [32].

Considering the need for continuous at-home monitoring in light of the prevalence of chronic conditions to ensure the empowerment and engagement of frail individuals, as well as in view of the need shown by some clinicians for knowledge regarding off-the-shelf technologies, we conducted a systematic literature review of research articles on wearable devices used for the continuous remote clinical monitoring of motor symptoms and physical activity [33,34] at home in individuals with disabilities related to known clinical conditions (musculoskeletal, neurological. or rheumatologic) or age-related conditions to provide insight into the following: the types of population and devices used, the purposes of use, and validity.

## 2. Materials and Methods

### 2.1. Search Protocol

We conducted a systematic search across three databases (PubMed, Scopus, and Web of Science) following the Preferred Reporting Items for Systematic Reviews and Meta-Analyses (PRISMA) guidelines, extracting papers published between 1 January 2013 and 19 July 2023. We chose this timeframe to include state-of-the-art evidence. Moreover, we hand-searched the references at the bottom of each included review to spot other reviews. We used the terms “wearable”, “monitoring”, “motor”, and “at home” as primary keywords and formulated the search strings following the established protocols and syntax conventions of each database consulted (Appendix A in Appendix A). The protocol was registered in PROSPERO with the identifier CRD42024604273.

### 2.2. Inclusion and Exclusion Criteria

We included (a) primary studies, (b) published in English, (c) focusing on the use of wearable devices for telemonitoring motor activity at home, (d) continuously for at least 24 h, (e) in patients with orthopedic, neurological, and/or rheumatologic diseases.

Studies were excluded if they (a) reported on single cases; (b) provided only unpublished data; (c) concerned healthy people or people in non-healthcare settings (e.g., frail elderly); (d) used prototype devices; or (e) restricted the observation to clinical or laboratory settings.

Based on the eligibility criteria, two pairs of reviewers (MC, EA, GF, NB) independently conducted the selection of studies. Conflicts were solved by one senior author (MGC).

### 2.3. Data Extraction

After consensus was reached regarding the included studies, three authors (AP, GF, and NB) extracted data about participants (total and subgroup numbers in case of controlled studies), study design, health condition, wearable devices, monitored functions/behaviors, monitoring duration, primary endpoints, results, effect sizes, ethical approval, and competing interests. The wearable devices were grouped by sensor description (sensory type, number of sensors used) and brand name (see Appendix A in Appendix A for a data extraction spreadsheet). Studies were grouped by levels of evidence based on the Oxford Centre for Evidence-Based Medicine (OCEBM) rules [35].

### 2.4. Quality Appraisal

The quality appraisal of studies was based on criteria adapted from “Standards for Reporting Diagnostic Accuracy Studies” (STARD) criteria [36], tailored to the main aims of this review. (See Appendix A in Appendix A for the quality appraisal.)

## 3. Results

### 3.1. Study Selection and Quality Appraisal

Of the 2984 studies analyzed by title and abstract, 108 progressed to the full-text analysis.

Four were excluded due to the unavailability of the full text. Out of the remaining 104, 72 were included in this systematic review. Figure 1 shows the PRISMA flowchart. Given the considerable heterogeneity in study tools, scope, design, and outcome measures, conducting a meta-analysis was deemed inappropriate, so the results were synthesized and presented using a narrative approach. The participants in the selected studies amounted to 7949 people, of which 5727 had pathological conditions (54 of them enrolled in a control group), and 2222 were healthy subjects (all in a control group). The study sample sizes ranged from a few units (*n* = 4) [33,34] to a few thousand (*n* = 4126) [28]. Mean, median, and modal sample size values were 110, 30, and 12, respectively. Six studies on Parkinson’s disease (PD) [28,37,38,39,40,41], one on Huntington’s disease [42], and one on multiple sclerosis [43] enrolled more than 90 subjects.

Based on the OCEBM rules, four studies were Level 2 (randomized controlled trials), 32 were Level 3 (2 non-randomized controlled trials and 30 cohort studies), and 36 were Level 4 (case series). The quality appraisal revealed that the majority of the selected studies clearly stated the key elements, including eligibility criteria (65, 90%), the rationale for choosing the reference standard (when alternatives were available) (61, 85%), the definition and rationale for test positivity cut-offs or index test result categories, distinguishing pre-specified from exploratory (41, 57%), methods for estimating or comparing measures of diagnostic accuracy (59, 82%), the baseline demographic and clinical characteristics of participants (65, 90%), the distribution of the severity of the disease in those with the target condition (66, 92%), and estimates of diagnostic accuracy with precision (such as 95% confidence intervals) (62, 89%). However, only a few studies reported whether participants formed a consecutive, random, or convenience series (7, 10%); how missing data on the index test and reference standard were handled (4, 5.5%); the intended sample size and how it was determined (1, 1.4%); the flow of participants using a diagram (8, 11%); any adverse events from performing the index test or using the reference standard (1, 1.4%) (Appendix A in Appendix A).

Out of the 72 included articles, 61 (85%) reported having received ethical approval, and 64 (89%) disclosed competing interests. Among the latter, 28 (44%) reported a competing interest.

### 3.2. Geographic Distribution

All studies were conducted in high-income countries, the majority (29, 40%) in the United States or Europe (34, 47%). The remaining studies were conducted in Canada (5, 7%), Asian countries (3, 4%), and Australia (1, 2%).

### 3.3. Health Conditions

A total of 86% (*n* = 61) of the studies include subjects suffering from neurological diseases or pathological conditions inherent in nervous system damage. A total of 35 works concern subjects with PD [28,34,37,38,39,40,41,44,45,46,47,48,49,50,51,52,53,54,55,56,57,58,59,60,61,62,63,64,65,66,67,68,69,70], 6 include subjects with cerebral stroke [71,72,73,74,75,76], 4 with multiple sclerosis [43,77,78,79], and 4 with Huntington’s disease [42,80,81,82]. One study includes age-related conditions such as the loss of balance in an elderly subject [83], while 11% (*n* = 8) of the studies concern subjects with orthopedic conditions, of which six include amputees of the lower limb [33,84,85,86,87,88]. Finally, two studies are related to rheumatic pathologies, osteoarthritis [89] and rheumatoid arthritis, osteoarthritis, or psoriatic arthritis [90]. A summary of the population and the sample size can be seen in Table 1.

### 3.4. Sensor Types

Various sensors were utilized in the included studies: accelerometers [34,43,45,48,53,57,60,61,64,65,68,70,71,72,73,74,75,77,78,81,82,85,87,88,90,91,92,93,94,95,96,97,98]; accelerometers and gyroscopes [37,38,39,40,42,44,46,47,49,50,52,55,56,59,67,69,76,80,84,99,100]; accelerometers and barometers [28]; accelerometers and electromyographs [62,101]; combinations of accelerometers, gyroscopes, and magnetometers [41,51,54,58,63,66,79,83,89,102]; and combinations of accelerometers, gyroscopes, magnetometers, and barometers [103,104]. Most of the devices used are commercially available and validated for remote movement monitoring. In such cases, data are processed using proprietary pre-installed algorithms, which provide information on the gait cycle, movement quantity, and posture maintained throughout the observation period. Thirteen studies used motion sensors integrated into smartphones [38,39,42,44,59,69,80] or commercial smartwatches [37,41,43,71,85,86].

Eight of these studies [38,39,41,42,44,59,69,80] exploited purposely developed apps (RocheHD Monitoring app version 1, FOX wearable companion app v2.x, Encephalog Home app, GaitReminder™ app, Brain Baseline app, two customized apps), using specific algorithms to process kinematic information from the sensors. Finally, eight studies [55,61,72,73,76,84,89,98] used patented sensors developed by the research group and not commercially available or validated.

A list of sensors and their features can be found in Table 2.

Patients wore sensors in different configurations and quantities, with the most frequently targeted body regions being the wrists (*n* = 30), lower back (*n* = 12), chest (*n* = 7), arm (*n* = 6), and foot (*n* = 5). Some sensors had less conventional applications, such as prosthetic pylons, shoes, or wheelchairs, often patented by the research group and not commercially available. The number of sensors used per study varied from 1 (*n* = 30) to 6 (*n* = 1).

### 3.5. Monitoring: Timelines and Targets

This review included studies that utilized sensors for no less than 24 h. This duration was observed in three of the analyzed studies [62,79,91], contrasting with the maximum period of 2.5 years in the study by De Lima et al. [28]. One week is the most frequently used monitoring period, common to 20 studies. Most studies (*n* = 54) have a monitoring period of less than or equal to 2 weeks. On average, the monitoring time is 46.5 days (SD = 145.346), while the mode and the median values are 7.

The studies involved in this review analyze several variables, namely the quantity of movement (QM), fall event/risk (FE), turning events (TEs), gait parameters (GPs), motor symptoms (MSs), and their correlation with clinician-assessed or self-reported scales/indexes (SC). Figure 2 matches information about the number of sensors applied to the patients, the parameters analyzed (OT stands for “other”), and the number of studies (green circle) involved in this matching. Figure 3 shows the number of sensors patients wear in different health conditions and how much this matching is represented in this review.

### 3.6. Monitoring System Feasibility

A total of 68% of the studies (*n* = 49) aim to demonstrate the feasibility of using systems to assess patients’ movement and behaviors remotely. Feasibility is expressed in acceptability/compliance, accuracy/reliability, and system validity. Eight studies [34,54,55,61,70,75,89,98] investigated patients’ reported acceptability of the used systems. These parameters were evaluated through interviews or questionnaires to define scores, indexes, or levels of satisfaction. The satisfaction questionnaires were self-completed by users or administered by the clinician in person or via telephone. The results of the questionnaires suggest a positive trend in device acceptability, particularly regarding comfort [54], ease of use [34], non-invasiveness, system quality [61], and comfort in remote communication with clinicians [98]. Eight studies investigated patients’ compliance, expressed through two parameters: (i) the amount of time the sensors were used during the day and (ii) the percentage values of use over the total experimentation time required. Studies report percentage values ranging from 56% [90] to 95.7% [60], calculated based on different parameters. The percentage values of compliance are not associated with the number of sensors used.

Articles describing health conditions related to the percentage of adherence/compliance and the number of sensors used are summarized in Table 3.

Thirty-two (44%) studies reported the validation of sensors or systems by comparing the collected data with an existing gold standard, a self-reported diary, or clinician judgment (Table 4). They provide (1) specificity, sensitivity, accuracy, and precision; (2) test–retest reliability associated with the intraclass correlation coefficient (ICC); and (3) predictive positive/negative value (PPV/NPV). The mean sensitivity and specificity are high (89.2% and 77.8%, respectively). The mean accuracy is 87%, as provided by four studies [33,44,52,84]. Other papers indicate that the systems used are generally accurate and reliable. No articles explicitly aimed to investigate the safety of the systems used. Only one [89] paper states that no adverse events occurred.

See Appendix A in Appendix A for data extraction.

## 4. Discussion

This systematic review investigated different techniques, tools, objectives, and clinical contexts related to the remote continuous monitoring of motor symptoms and physical activity at home across various health conditions, i.e., musculoskeletal conditions, neurological conditions, rheumatologic diseases, or aging-related disorders. We reported the feasibility of systems and architectures, focusing on their acceptability, reliability, and clinical usability, and conducted a critical assessment of the included studies. Our findings offer an overview of the current state of wearable sensor applications for motor assessment at home, providing a starting point for practical implementation in real-world care settings and for the design of further research projects. We included 72 articles out of the 2984 initially identified documents. The quality of evidence showed that only 50% of studies had a controlled design, and four were randomized. The remaining 50% were descriptive reports. Some appraisal issues emerged in the analysis: a few studies reported whether participants formed a consecutive, random, or convenience series or how missing data on the index test and reference standard were handled; the intended sample size and how it was determined; the flow of participants using a diagram; and any adverse events from performing the index test or using the reference standard. Despite these limitations, the included studies were generally considered reliable, as their design choices were aligned with the study scopes, with the majority being feasibility studies accounting for 68% (*n* = 49) of the total. Therefore, it emerged that home-based continuous monitoring had been applied to study neurologic, aging-related, orthopedic, and rheumatologic diseases. Interestingly, the most frequently monitored conditions are those affected by neurological disorders. PD (*N* = 35 studies on 4646 patients) is the most addressed condition, followed by Huntington’s chorea (*N* = 4 studies on 263 patients), multiple sclerosis (*N* = 4 studies on 189 patients), and cerebral stroke (*N* = 6 studies on 107 patients). Diagnosing or monitoring specific disease-related motor features was the most frequent scope of these studies, while the global motor activity and gait parameter monitoring recurred as goals in studies conducted on patients with stroke or multiple sclerosis. Although stroke remains the leading cause of disability in the adult population, and multiple sclerosis is not uncommon [1], movement disorders, such as PD, are widespread, with a steadily rising prevalence and incidence. These evolutive conditions are disabling and present several characteristics that make them particularly well-suited for remote assessment through technology: the need for continuous monitoring associated with mobility limitation, the presence of episodic symptoms (freezing of gait, falls, tremors, dyskinesias), and the presence of highly specific motor disturbance. Different systems offer bilateral monitoring and include smartphone applications where patients can log their medication intake and overall health.

Physicians, in turn, may receive indications to change or adjust the patient’s medication or rehabilitation. Some smartphone apps can also be tailored to detect PD symptoms, such as bradykinesia, tremors, vocal abnormalities, or changes in facial expression. Such tools can be applied for the early detection of PD in the general population and monitoring disease progression in known PD patients. It is notable that most PD patients can use these digital solutions. In today’s world, wearable sensors and telemedicine provide a promising new approach to managing PD, offering easy-to-use tools accessible to most people [106].

In non-neurological conditions, except for elderly care and preventive health approaches, remote home-based monitoring using wearable sensors is less common. This is often because the conditions being treated are acute rather than chronic, with motor symptoms that are less fluctuating, episodic, or distinctive and more predictable than in movement neurological disorders.

Most of the studies that included wearable sensors utilized medical or commercial devices equipped with proprietary software developed by the parent company. This software translates kinematic data into clinical information and is typically designed to extract parameters related to patients’ motion and gait characteristics. Additionally, various studies have attempted to develop algorithms using raw data from these devices to provide further insights, such as detecting motor symptoms specific to the condition (tremors, bradykinesia, freezing of gait, dyskinesias), assessing fear or risk of falling, and evaluating both the quality and quantity of directional changes.

This review highlights a certain heterogeneity in the types of devices used. We can distinguish between (i) multisensor devices, such as IMUs constituted by accelerometers, gyroscopes, and magnetometers (16%) rather than more simple devices constituted by one (48%) or two (36%) of these tools, and (ii) medical devices rather than commercial smartphone- or smartwatch (18%)-based devices. Most of the medical devices used are commercially available and have been validated for remote movement monitoring. In a digital world that is rapidly evolving, we underlined the need to focus on both the available evidence related to smartphones and the new technologies and/or devices that might facilitate the use of wearable sensors for motor monitoring.

Different sensor characteristics may affect their usability, in particular, the monitoring timelines, number of worn devices, and the site of application. The monitoring durations ranged from a couple of days to more than two weeks, with a median of 7 days and a range of 2 days–2, 5 years. Among the commercial medical devices, the most commonly used for periods of less than 3 days were the DynaPort (DynaPort Hybrid, McRoberts, The Hague, The Netherlands) and BiostampRC (MC10 Inc., Lexington, MA, USA). For monitoring periods between one and two weeks, the Opal wearable sensor (APDM, Inc., Portland, OR, USA) was the most frequently used. No clear and specific association between the selected devices and the monitoring duration was found. However, commercial smartphones and smartwatches equipped with either commercial apps or apps developed by research groups were more frequently used for longer than a few days. This preference is likely due to their greater acceptability (as they are not stigmatizing medical devices) and ease of use, ensuring comfort, adherence, and compliance from the participants [38,107].

The number of sensors used ranged from one to six, with 63% of studies using only one device. The choice may depend on the monitoring goals/targets. For the assessment of movement quantity or gait parameters, authors more frequently chose to use one or two devices. Only when the purpose of monitoring was to find correlations between complex motor behavior and clinician-assessed or self-reported scales/indexes did researchers prefer to use more than one device [19].

Other study objectives, in addition to those previously mentioned, included the detection of fall events or fall risk, the identification of specific motor disorders such as tremors or dyskinesia, and the analysis of gait-specific events like turning. The most frequently targeted aspect was the monitoring of movement quantity (43%), followed by gait parameters (26%) and specific motor symptoms (26%). A potential correlation may exist between the population studied and the monitoring objectives. For instance, monitoring was conducted for the freezing of gait, tremor, and dyskinesia in individuals with PD; fall risk was assessed in older adults; and arm movements were observed in those with shoulder conditions.

Most of the included papers aimed to demonstrate the feasibility related to either (i) developing new algorithms extracting raw data from already marketed sensors, (ii) using already validated sensors or systems to analyze motor behaviors, or (iii) designing and developing new sensors and related systems in experimental settings. The importance of investigating the research question in the current study is supported by a widespread socioeconomic need for care across various geographical regions. We emphasized that remote continuous monitoring provides a detailed and reliable overview of the health conditions and social behavior of the subjects analyzed. Additionally, it helps reduce healthcare costs and minimizes travel times for patients in underserved areas.

Concerning the analysis of the usability or acceptability of technology, forty-nine studies (68%) analyzed the feasibility of the wearable motor monitoring system; eight studies (11%) analyzed compliance, which overall exceeded 70%, supporting the hypothesis that patients were inclined to accept monitoring systems (108). Similarly, acceptability was reported by eight studies, which showed good acceptability estimated through questionnaires. We, therefore, believe that several groups of authors have reached a consensus in stating a reliable level of acceptability. The comparison between these systems and clinical evaluations in the laboratory using gold standard procedures (with already validated systems and with self-reported information) revealed the high reliability of wearable sensors in monitoring.

This evidence opens the possibility of considering the administration of remote motor monitoring at home to integrate and complete the clinical ambulatory assessment for the characteristics that we cannot observe in the laboratory because they are episodic, attention-dependent, or too lengthy to record. Moreover, different studies underlined the reliability of non-medical and more cost-effective devices (i.e., smartwatches). However, we stressed that this is the first piece of evidence, and further experimental results are required across different contexts, populations, sensors, and monitoring methodologies (e.g., length of monitoring, number of sensors employed).

It is of note that the use of a big amount of data deriving from the use of sensors at home could represent a real value, and if combined with artificial intelligence, in the future, it could help open up unexplored scenarios such as (a) an efficient “continuum of care” (transfer of patient data from one rehabilitation setting to another); (b) the early prediction of pathological conditions; (c) and efficient interoperability between systems (those involved in clinical setting, research, and healthcare management) [108]. Before the current study, other authors addressed similar research questions. Specifically, among published secondary studies on our research question, one systematic review focused on records published between 1 January 2015 and 24 June 2020 on motor assessment through digital devices [109]. However, our review extends the scope by including studies published until the first half of 2023.

Most of the existing literature reviews about movement monitoring tend to focus on specific health conditions, such as rheumatic and musculoskeletal diseases [110], dementia or mild cognitive impairment [111], PD [112], and multiple sclerosis [113].

Moreover, a key point of our review is the distinction between laboratory and everyday scenarios, which was not addressed in other reviews [109,114]. In addition, we focused on different targets touching on a wide range of clinical outcomes, whereas Patel et al. concentrated on a specific target: the risk of falls [19].

### Limitation

The limitations of our review may stem from including only peer-reviewed studies published in English, which predominantly reflect experiences from the Western, Educated, Industrialized, Rich, and Democratic (WEIRD) contexts [115,116].

The scarcity of high-quality evidence in the literature, such as randomized trials (comprising 5.6% of the total) or non-randomized controlled trials and cohort studies (44%) along with the typical objectives of studies on wearable systems in home settings, led us to include experimental results from diverse research designs. While this approach may impact the overall quality of the evidence, it enables a more comprehensive exploration of the research question across various objectives and health conditions [117].

Possibly, as highlighted by Lin et al. (2025) [118], who analyzed 1572 publications using a scientometric approach to define trends and hotspots in the field of wearable devices for sleep science, bibliometric methods offer powerful tools for clarifying the niche and scope of a research domain. In our systematic review, we addressed similar objectives by focusing on continuous home-based motor monitoring through wearable sensors, including comorbid conditions such as neurodegenerative, rheumatologic, and musculoskeletal diseases. We agree that future work could benefit from further expanding the dataset and more systematically exploring the role of wearable technologies in combined pathologies, which are indeed highly relevant in real-world clinical contexts [118].

The use of wearable systems in clinical settings, mainly when employed at home, raises several ethical considerations that were not always assessed or considered. First, privacy and data security issues are paramount, as these devices often collect sensitive health information. Ensuring that patient data is protected from unauthorized access or misuse is critical, especially in the context of remote monitoring. Additionally, the accuracy and reliability of data provided by wearable devices must be carefully assessed to avoid potential harm from incorrect diagnoses or treatment decisions based on inaccurate readings. There is also the ethical concern of informed consent, as patients should fully understand how their data will be used, stored, and shared; we underlined that this piece of information is not always collected, and we did report this information.

Finally, equity in access to such technologies must be considered, as not all patients may have the resources or technological literacy to use wearable systems effectively. Clinicians must be mindful of these ethical issues to ensure that wearable systems enhance patient care without compromising patient rights or well-being [119,120,121,122,123].

## 5. Conclusions

This review highlights the potential of wearable sensors for the remote, continuous monitoring of motor symptoms and physical activity in home settings across various health conditions, particularly neurological disorders like PD. The findings demonstrate the feasibility, usability, and reliability of these systems, with most studies reporting high patient compliance and acceptance. Wearable devices, including medical-grade and commercial options, were widely used, though methodologies and monitoring durations varied significantly. Neurological conditions were the primary focus, while non-neurological applications were less common. Despite the promising results, evidence quality was limited, with only 50% of studies using controlled designs and a few randomized trials. Ethical concerns such as data privacy, informed consent, and equitable access remain critical and require greater attention. Future research should prioritize high-quality trials, standardize methodologies, and address cost-effectiveness to broaden accessibility. This review underscores the potential of wearable sensors to enhance real-world patient care while identifying areas requiring further development. Future advancements, including artificial intelligence integration, could enhance data interpretation and predictive capabilities, making remote monitoring more personalized and efficient. Finally, there is a prominent request from the scientific community [116] to collect data on science in non-WEIRD countries, specifically the Call for Action of the Rehabilitation 2030 of the WHO.

## Figures and Tables

**Figure 1 sensors-25-04889-f001:**
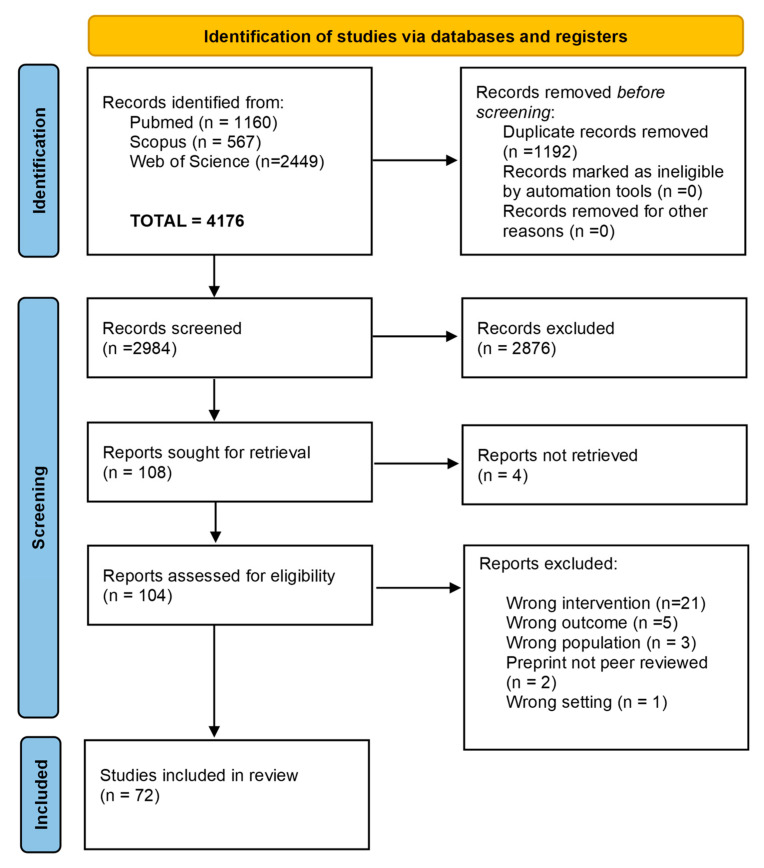
Flowchart of screening procedure according to PRISMA guidelines.

**Figure 2 sensors-25-04889-f002:**
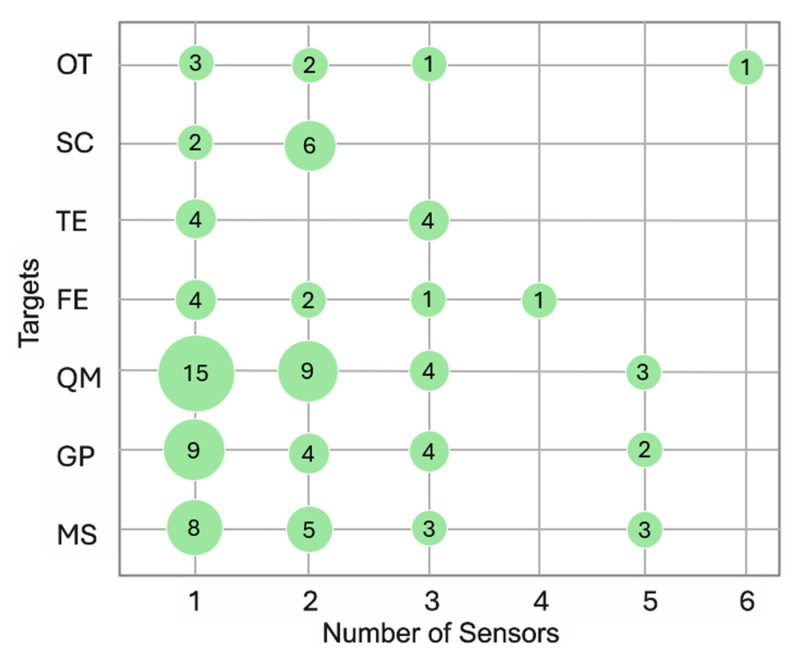
Information about the number of sensors applied to the patients, the parameters analyzed (quantity of movement (QM), fall event/risk (FE), turning events (TEs), gait parameters (GPs), motor symptoms (MSs), clinician-assessed or self-reported scales/indexes (SC), other (OT)), and the number of studies involved in this matching. Legend: green circle = studies involved; size of the circle = proportional to the number of the studies involved.

**Figure 3 sensors-25-04889-f003:**
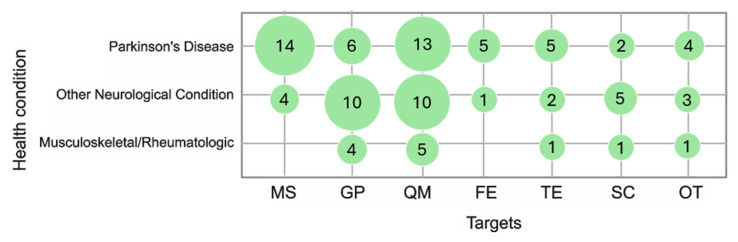
Targets (quantity of movement (QM), fall event/risk (FE), turning events (TEs), gait parameters (GPs), motor symptoms (MSs), clinician-assessed or self-reported scales/indexes (SC), other (OT)) analyzed in the different health conditions and how much this matching is represented in this review. Legend: green circle = articles describing the related pathological condition; size of the circle = proportional to the number of the articles involved.

**Table 1 sensors-25-04889-t001:** Overview of included studies and subjects by health condition, grouped by clinical category.

Health Condition	Studies (n°)	Subjects (n°)
Parkinson’s Disease	35	4646
Stroke	6	107
Multiple Sclerosis	4	189
Huntington Disease	4	263
Ataxia	3	62
Alzheimer’s Disease	2	28
Other	7	227
**Total Neurologic Conditions**	**61**	**5522**
Loss of Balance	1	5
**Total Aging-related Conditions**	**1**	**5**
Amputee	6	94
Polytrauma	1	48
Degenerative Facet Osteoarthropathy	1	8
**Total Musculoskeletal Conditions**	**8**	**150**
Osteoarthritis	1	20
Osteoarthritis–rheumatoid arthritis–psoriatic arthritis	1	30
**Total Rheumatologic Conditions**	**2**	**50**
Subjects with Pathological Condition		5727
Healthy Subjects (Control Groups)		2222
**Total**	**72**	**7949**

**Table 2 sensors-25-04889-t002:** Summary of sensors: This table details the sensors’ hardware, commercial names, and additional applications or data loggers’ name if required. The final three columns specify each device’s commercial or medical validation status and indicate whether it holds CE certification. Items marked ’n.a.’ denote products that are no longer commercially available.

Hardware	Device’s Name	Additional App or Data Logger	References	Commercial Device	Medical Device	CE Certification
Tri-axial accelerometer	BiostampRC (MC10 Inc., Lexington, KY, USA)	Not needed	[70,77,81,91]	yes	yes	yes
	Axivity AX-3 (Axivity Ltd., Newcastle upon Tyne, UK)		[57,78,94]	yes	no	no
	ActiGraph GT3X o GT9X (ActiGraph LLC, Pensacola, FL, USA)		[88,90,92]	yes	no	no
	GeneActiv (ActivInsights Ltd., Kimbolton, UK)		[45,95,96]	yes	yes	no
	RehaGait (Hasomed GmbH, Magdeburg, Germany)		[64,65]	yes	yes	no
	PAMSys (BioSensics, London, UK)		[82,97]	yes	no	no
	ActivPal (PAL Technologies Ltd., Glasgow, UK)		[75,87]	yes	no	no
	PKG (Parkinson KinetiGraph) (Global Kinetics/PKG Health, Melbourne, Australia)		[68]	no	yes	no
	REMPARK (CETpD—Universitat Politècnica de Catalunya, Barcelona, Spain)		[53]	no	no	no
	Shimmer (Shimmer Research, Dublin, Ireland)		[48]	yes	no	yes
	Kinesia (Great Lakes NeuroTechnologies Inc., Cleveland, OH, USA)		[60]	yes	yes	no
	Actibelt RCT2 (Trium Analysis GmbH, Munich, Germany)		[93]	yes	yes	no
	Actical (Philips Respironics, Bend, OR, USA)		[74]	yes	no	no
	ITEX gloves (University of Rhode Island—Wearable Biosensing Lab, Kingston, UK)		[34]	no	no	no
	Not specified		[61,72,73,98]			
**Tri-axial** **accelerometer–gyroscope**	Dynaport Hybrid (McRoberts B.V., The Hague, The Netherlands)	Not needed	[40,46,67]	yes	yes	yes
	Physilog (MindMaze Assessments, Lausanne, Switzerland)		[99,100]	yes	no	yes
	Mobile GaitLab (Portabiles HealthCare Technologies GmbH, Erlangen, Germany)		[47,52]	yes	yes	yes
	MOX5 (Maastricht Instruments/Instrument Development Engineering & Evaluation department, Maastricht, The Netherlands)		[49]	yes	no	yes
	PD Monitor (PD Neurotechnology Ltd., London, UK)		[105]	yes	yes	yes
	STAT-ON (Sense4Care, Barcelona, Spain)		[50]	no	yes	yes
	Gait Tutor System (MHealth Technologies, Bologna, Italy)		[56]	yes	yes	no
	Not specified		[55,76,84]			
**Tri-axial** **accelerometer–barometer**	PERS-phylips (Philips Lifeline, Cambridge, MA, USA)	Not needed	[28]	yes	yes	yes
**Tri-axial** **accelerometer–surface electromyography**	FarosEMG (Bittium Biosignals Ltd., Kuopio, Finland)	Not needed	[62,101]	n.a.	n.a.	n.a.
**Tri-axial** **accelerometer–gyroscope–magnetometer–barometer**	ReSense (Rehabilitation Engineering Lab, ETH Zurich, Zurich, Switzerland)	Not needed	[103]	no	no	no
**Accelerometer–gyroscope–magnetometer**	Opal APDM, Inc (APDM Wearable Technologies, Portland, OR, USA)	Not needed	[51,58,63,66,79,83,102]	yes	no	no
	GaitAssist (Swiss Federal Institute of Technology Zurich, Zurich, Switzerland)		[54]	no	no	no
	ActiMyo (Institute of Myology & Sysnav partnership, Paris, France)		[104]	no	yes	yes
	Not specified		[89]			
**Coil antenna**	WAFER (Department of Bioengineering, University of Washington; Seattle, WA, USA)	ECHO	[33]	no	no	no
**Smartphone sensors** **(accelerometer, gyroscope, GPS)**	Verily study watch (Verily Life Sciences LLC, Dallas, TX, USA)	Not needed	[37]	yes	yes	yes
	Samsung Galaxy J7 (Samsung Electronics Co., Suwon, Republic of Korea)	ROCHE HD	[42]	yes	no	yes
	iPod touch 4th generation (Apple Inc., Cupertino, CA, USA)	Gait Reminder	[59]	yes	no	yes
	iPhone 5 or 6 (Apple Inc., Cupertino, CA, USA)	Customized App	[44]	yes	no	yes
	iPhone 10 or 11 (Apple Inc., Cupertino, CA, USA)	Brain Baseline	[41]	yes	no	yes
	Android Phone (not specified)	Encephalog Home FOX wearable companion Customized app	[38,39,69,80]			
**Smartwatch sensors (accelerometer, gyroscope, GPS)**	Moto G360 (Motorola Mobility LLC, Chicago, IL, USA)	ROCHE HD	[42]	yes	no	yes
	Stepwatch Activity Monitor (Modus Health LLC, Washington, DC, USA)	Not needed	[86]	n.a.	n.a.	n.a.
	Apple Watch 4 or 5 (Apple Inc., Cupertino, CA, USA)	Brain Baseline	[41]	yes	no	yes
	FitBit (Fitbit Inc., San Francisco, CA, USA)	Not needed	[43,71,85]	yes	no	yes
	Pebble Smartwatch (Pebble Technology Corp., Palo Alto, CA, USA)	Fox wearable companion	[38,39]	yes	no	Yes

**Table 3 sensors-25-04889-t003:** Patients’ compliance: health condition and number of sensors used, related to percentage values of compliance/adherence.

Health Condition	% Adherence /Compliance	Number of Sensors Used	References
Parkinson’s disease	71	5	[105]
	68	2	[38]
	59	1	[37]
	94	3	[49]
	96	1	[60]
Stroke	91	1	[76]
Cerebral palsy	Not expressed	1	[93]
Osteoarthritis–rheumatoid arthritis–psoriatic arthritis	56	2	[90]

**Table 4 sensors-25-04889-t004:** System validation: articles involved in system validations expressed in comparison with gold standard algorithms, patient-reported outcomes, and clinical assessment. For each type of comparison. sensitivity, specificity, accuracy, precision, and ICC are shown.

Validation Method	References	Sensitivity	Specificity	Accuracy	Precision	ICC
Gold standard	[43,44,52,84,93]	n.a.	n.a.	82.5%	n.a.	0.76
Patient-reported outcome	[53,80,83,90,101]	92.6%	97.6%	97.6%	96.4%	0.70
Clinical assessment	[33,34,37,42,44,45,46,47,48,49,50,51,71,72,77,78,80,81,89,91,92,99,101,102,103]	82.5%	63.5%	70.0%	n.a.	0.72

## Data Availability

The original contributions presented in this study are included in the article. Further inquiries can be directed to the corresponding author.

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
