# Peer review of "Continuous Movement Monitoring at Home Through Wearable Devices: A Systematic Review"

_sensors, 2025, doi:10.3390/s25164889_

Round 1

Reviewer 1 Report

Comments and Suggestions for Authors
  • A complex bibliographical research, with many important aspects addressed.
  • For a clear exemplification of the niche of the analysis field related to wearable sensors, it would be beneficial to use a scientometric analysis that could include many more research papers.
  • For a future continuation or even improvement of this article, it would be useful to identify and briefly mention the use of wearable sensors for combined pathologies (perhaps much more common in reality).

Author Response

Reviewer’s Comment

A complex bibliographical research, with many important aspects addressed.

For a clear exemplification of the niche of the analysis field related to wearable sensors, it would be beneficial to use a scientometric analysis that could include many more research papers.

For a future continuation or even improvement of this article, it would be useful to identify and briefly mention the use of wearable sensors for combined pathologies (perhaps much more common in reality).

Answer: We are grateful to the reviewer for the very insightful comments. We agree that future developments should focus on the information recorded from wearable sensors applied in cases with multiple and complex disabilities. We added this comment in the Discussion – subsection Limitations

Reviewer 2 Report

Comments and Suggestions for Authors

This manuscript presents a review of wearable systems designed for continuous movement monitoring in home environments. The authors analyze literature published over a 7-month period, aiming to synthesize recent developments and trends in this area. I think that the topic is relevant, especially given the growing emphasis given to the continuous monitoring of the human movement outside a controlled lab.

While the review addresses an important topic, the decision to limit the analysis to a 7-month period is not sufficiently justified. The authors do not explain why this short timeframe is meaningful, nor do they provide compelling evidence of a major shift or breakthrough that would warrant such a focused review.

Furthermore, the paper does not offer enough critical analysis or synthesis of the related literature — particularly considering the substantial research conducted in this area over the past decade or more — to make it a useful resource for the community. If the authors wish to resubmit, they should consider expanding the scope to cover a longer period and provide a deeper, more critical discussion of trends, gaps, and technological developments in the field.

Author Response

Reviewer’s Comment

This manuscript presents a review of wearable systems designed for continuous movement monitoring in home environments. The authors analyze literature published over a 7-month period,  aiming to synthesize recent developments and trends in this area. I think that the topic is relevant, especially given the growing emphasis given to the continuous monitoring of the human movement outside a controlled lab.

While the review addresses an important topic, the decision to limit the analysis to a 7-month period is not sufficiently justified. The authors do not explain why this short timeframe is meaningful, nor do they provide compelling evidence of a major shift or breakthrough that would warrant such a focused review.

Furthermore, the paper does not offer enough critical analysis or synthesis of the related literature — particularly considering the substantial research conducted in this area over the past decade or more — to make it a useful resource for the community. If the authors wish to resubmit, they should consider expanding the scope to cover a longer period and provide a deeper, more critical discussion of trends, gaps, and technological developments in the field.

Answer: We thank the reviewer for the comment. However, we would like to point out that there has been a misunderstanding regarding the period covered by our literature search. In the Method Section we have clearly stated that the search focused on papers published from January 1,, 2013 to July 19, 2023, thus covering a period longer than 10 years and including recent literature.

As the reviewer correctly argues, the inclusion of papers published over the last decade meets the need for providing critical insight into recent technology development.

Reviewer 3 Report

Comments and Suggestions for Authors

This manuscript provides a review of wearable sensors for continuous motor monitoring in home settings, with a particular focus on their objectives, sensor types, feasibility, and effectiveness in managing neurological, musculoskeletal, or rheumatologic conditions. I would like to offer the following suggestions to help improve the quality of the manuscript:

(1) Typically, 4–6 keywords are recommended.

(2) The captions for figures and tables should be clear, precise, and informative to improve readability. For example, the caption of Table 1 reads “populations’ summary”—it would be helpful to clarify: summary of what aspect of the populations?

(3) Is Figure 3 created by the authors? If so, please ensure its font style and overall design are consistent with Figure 1.

(4) The figure referenced around Line 259 is missing a proper caption. Please ensure all figures are clearly labeled and described.

(5) In Section 4 Discussion, the presence of a single subsection labeled 4.1 is confusing. What section does the preceding content belong to—should it be considered Section 4.0? Please revise the formatting to improve structural clarity.

(6) Since the authors have discussed certain limitations in Section 4.1, the manuscript would benefit from suggesting potential solutions or future directions to address these limitations.

Author Response

Reviewer’s Comments

Comment 1:

This manuscript provides a review of wearable sensors for continuous motor monitoring in home settings, with a particular focus on their objectives, sensor types, feasibility, and effectiveness in managing neurological, musculoskeletal, or rheumatologic conditions. I would like to offer the following suggestions to help improve the quality of the manuscript:

  • Typically, 4–6 keywords are recommended.

Response C (1): Thank You very much for the suggestion: we select the follow six: Wearable sensors; Home-based monitoring; Motor symptoms; Inertial Measurement Units (IMUs); Remote assessment; Digital health.

Comment 2:

The captions for figures and tables should be clear, precise, and informative to improve readability. For example, the caption of Table 1 reads “populations’ summary”—it would be helpful to clarify: summary of what aspect of the populations?

Response C (2): Thank You very much for the suggestion: we modify the following table and caption in order to improve their readability:

Table 1. Populations’ summary. Overview of included studies and subjects by health condition, grouped by clinical category

Comment 3:

Is Figure 3 created by the authors? If so, please ensure its font style and overall design are consistent with Figure 1.

Response C (3): Thank You very much for the suggestion; I confirm that the figure 3 was created by one of our authors, so we have modified the figures 2 and 3 as You requested.

Comment 4:

The figure referenced around Line 259 is missing a proper caption. Please ensure all figures are clearly labeled and described.

Response C (4): The figure around the Line 259 is the third one (Figure 3). Its caption is above in the previous page (Line We linked better the Caption to the figure in the text.

Comment 5:

In Section 4 Discussion, the presence of a single subsection labeled 4.1 is confusing. What section does the preceding content belong to—should it be considered Section 4.0? Please revise the formatting to improve structural clarity.

Response C (5): Thank You very much to point out the question. We removed the point number in order to improve the readability.

Comment 6:

Since the authors have discussed certain limitations in Section 4.1, the manuscript would benefit from suggesting potential solutions or future directions to address these limitations.

Response C (6): Thank You very much, we agree and improved the Discussion sub-section Limitation some potential solutions, i.e. the scientometric analysis.

Round 2

Reviewer 2 Report

Comments and Suggestions for Authors

Thank you for the clarification. I acknowledge the time period specified in the methods section and agree that the literature search appropriately covers the past decade, including recent developments relevant to the topic. With this clarification, the concern is fully addressed.

Reviewer 3 Report

Comments and Suggestions for Authors

The manuscript has been improved after revision.